# A Digitally Enabled, Pharmacist service to detecT medicine harms in residential aged care (nursing home) (ADEPT): protocol for a feasibility study

Monique S Boord [1], Peter Brown,[2] Julian Soriano,[3,4] Tahlia Meola,[5] Dorothea Dumuid,[6] Rachel Milte,[7] Elizabeth E Roughead [8], Nigel H Lovell,[2] Helen Stone,[9] Joseph Whitehouse,[10] Jack L Janetzki,[1,5] Eyob Alemayehu Gebreyohannes,[8,11] Renly Lim [8]

For numbered affiliations see end of article.

**Correspondence to**
Dr Monique S Boord;
Monique.Boord@unisa.edu.au

## ABSTRACT

**Introduction** This feasibility study aims to develop and test a new model of practice in Australia using digital technologies to enable pharmacists to monitor early signs and symptoms of medicine-induced harms in residential aged care.

**Methods and analysis** Thirty residents will be recruited from an aged care facility in South Australia. The study will be conducted in two phases. In phase I, the study team will work with aged care software providers and developers of digital technologies (a wearable activity tracker and a sleep tracking sensor) to gather physical activity and sleep data, as well as medication and clinical data from the electronic medication management system and aged care clinical software. Data will be centralised into a cloud-based monitoring platform (TeleClinical Care (TCC)). The TCC will be used to create dashboards that will include longitudinal visualisations of changes in residents' health, function and medicine use over time. In phase II, the on-site pharmacist will use the centralised TCC platform to monitor each resident's medicine, clinical, physical activity and sleep data to identify signs of medicine-induced harms over a 12-week period.

A mixed methods process evaluation applying the RE-AIM (Reach, Effectiveness, Adoption, Implementation, Maintenance) evaluation framework will be used to assess the feasibility of the service. Outcome measures include service reach, changes in resident symptom scores (measured using the Edmonton Symptom Assessment System), number of medication adverse events detected, changes in physical activity and sleep, number of pharmacist recommendations provided, cost analysis and proportion of all pharmacists' recommendations implemented at 4-week, 8-week and 12-week postbaseline period.

**Ethics and dissemination** Ethical approval has been obtained from the University of South Australia's Human Research Ethics Committee (205098). Findings will be disseminated through published manuscripts, conference presentations and reporting to the study funder.

**Trial registration number** ACTRN12623000506695.

## STRENGTHS AND LIMITATIONS OF THIS STUDY

⇒ Combined various sources of health data into a single centralised platform for the residential aged care pharmacist to actively monitor medicine-induced harms.
⇒ Uses objective assessments of changes in physical activity and sleep to identify 'mild' medicine harms.
⇒ Due to recruiting from a single residential aged care facility in South Australia, findings may not generalise across all residents living in aged care.
⇒ Lack of compliance with use and wearing of digital technologies by residents is a potential limitation.

## INTRODUCTION

Approximately 8% of adults in Australia aged 65 years and older reside in aged-care facilities (nursing home),[1] with up to 63% receiving nine or more regular medications.[1] Medicine-related problems contribute to an estimated 250 000 hospital admissions in Australia every year, with an estimated annual cost of US$1.4 billion.[2] Medicine-related harms are of significant concern in residential aged care, with up to 27 adverse medicine events occurring per 100 residents per month.[3–5] In Australia, about one in five persons living in residential aged care experience an adverse medicine event every month; of which 83% were potentially preventable.[6] Despite the significant medicine-related harms, most Australian residential aged care facilities have limited capacity to provide clinical pharmacy services; existing services are limited to government-funded residential medication management review (comprehensive medication reviews performed by accredited pharmacists in residential aged care) and quality use of medicine services.[7]

The Australian Government, recognising the significance of medicine-related problems in aged care, committed US$350 million to improve medication safety in residential aged care through funding for pharmacists.[8] The government investment presents new opportunities for pharmacists to review residents for signs and symptoms of medicine harms more frequently. The present study builds on our recent randomised controlled trial which investigated the effectiveness of ongoing pharmacist assessments compared with usual care in reducing medicine-induced deterioration, frailty and adverse events across 39 residential aged care facilities in Australia.[9 10] Our study showed that one in five residents experienced a preventable adverse event (eg, falls, fractures) every month due to medication use.[9] The pharmacists identified and resolved medicine-related harms and served as an advocate for residents. Furthermore, significant differences in change in cognition from baseline were observed between intervention and control groups.[9] The service ensured pharmacist engagement with residents, rather than just reviewing medicines, which enabled the advocacy.[11]

Frequent monitoring of medicine-induced harms is important because medicines can cause mild harm (eg, sedation, poor balance) that often remains undetected, but can lead to serious adverse events in older people if left unmanaged. For example, commonly prescribed psychotropic medicines can cause sedation and poor cognition that may lead to injurious falls and fractures.[12] An increase of sedative load from two to four over a 12-month period has been associated with an average increase in daily sedentary behaviour of approximately 24 min[13]; an increase of 60 min/day of sedentary time is associated with a 19% increase in risk of death.[14] Detecting mild, cumulative effects of medicine-induced harms, including changes in physical activity, is likely to be important, however, this can be difficult and labour intensive to measure and monitor.

Harnessing digital technologies to enable active monitoring of medicine-related harms is one potential strategy to enabling proactive monitoring. For pharmacists to effectively monitor harms, they will not only need to review medicines, but also recognise and monitor early signs of medicine-induced harms such as changes in sleep, activity or cognition.[9 13] Digital technologies can measure changes in sleep, activity or cognition objectively, which may enhance effective communication about medicines-related harm across aged care residents' care teams, including to general practitioners (GPs), nurses and care staff as well as support the recommendations from pharmacists and their acceptance by the team and residents themselves. The TeleClinical Care (TCC) platform has been used successfully during the COVID-19 pandemic to safely manage >8000 COVID-19-positive patients as they self-isolated at home[15] and is being used in trials to manage cardiovascular disease and stroke.[16]

The ADEPT project aims to develop and implement a digitally enabled, evidence-based, pharmacist service built on TCC (TCC-ADEPT) to actively detect medicine harms in residential aged care. Specifically, we will:

1. determine the number and proportion of eligible residents who were willing to participate in the study;
2. determine changes in resident outcomes, including changes in symptoms as measured using the Edmonton Symptom Assessment System (ESAS),[17] physical activity and sleep (24-hour movement behaviour) and adverse events including falls (both injurious and non-injurious), fractures, delirium, faecal impaction, hospitalisation or emergency department presentation due to adverse medication events and medication incidents, predigital and postdigital pharmacist service;
3. determine the number and types of pharmacist recommendations that resulted following use of the TCC-ADEPT platform, and the number of pharmacist recommendations that were accepted and implemented by GPs and aged care staff;
4. explore the acceptability of the digital pharmacist service by residents and family members, aged care staff and GPs;
5. assess the costs associated with implementing the TCC-ADEPT platform and potential cost impacts to the health and aged care system in wider-spread roll out of the platform;
6. explore the residential aged care facility's intention to continue to implement the pharmacist service post-study funding and whether the service aligns to the facility's mission or sustainability of business model.

We hypothesise that the new pharmacist service is feasible and acceptable to aged care staff, residents and their GPs, and can enhance detection of early signs and symptoms of medicine harms, triggering review and action when required.

## Methods and analysis

### Study design
The ADEPT project will be conducted in two phases over a 12-month period. Phase I will include gathering data from digital technologies, medication and health records into the TCC platform. Digital technologies will include an activity tracker[18] and a sleep sensor.[19] A pre-post 12-week implementation study will constitute phase II of the project. The on-site pharmacist will monitor all participants using the centralised TCC-ADEPT platform, using data to inform resident reviews and to initiate conversations with residents, GPs and aged care staff on medicine, health and quality of life changes for residents. Recruitment and study implementation will occur from August 2023 until February 2024.

### Patient and public involvement
Aged care residents, representatives and health professionals were directly involved in the development or design of this study. Findings from interviews with

residents, carers and aged care staff from the Reducing Medicine-Induced Deterioration and Adverse Reactions (ReMInDAR) trial were used to inform the development and design of this study.[20]

## Cohort selection

Residents will be considered eligible to participate if they are a permanent resident living in the designated residential aged care facility. Exclusion criteria include receiving palliative or respite care.

## Screening and recruitment

Residents will be recruited from an aged care facility in South Australia. Promotional recruitment material (flyers) will be distributed onsite in common areas of the facility 1 month prior to study commencement. Participant information sheets will be provided to residents deemed potentially eligible by facility staff, and will be given to all residents willing to participate, informing them of the study details and pharmacist service.

Study staff will perform initial screening of residents' care record to identify potentially eligible residents. Screening logs will be kept monitoring residents' recruitment and to enable a description of the study population from which eligible residents have been enrolled. If the resident fulfils inclusion criteria based on the initial screening process, the on-site pharmacist will invite the resident to participate in the study and written informed consent will be obtained from the resident or their authorised representative (model consent form shown in online supplemental material).

## Baseline data collection

Once written consent is obtained, study staff will collect baseline data from the Resident Care Assessment Record held electronically in Leecare[21] and Medi-Map[22] software, which contains demographic details, comorbidities, medicine use, falls history and care provided within the aged care facility. Study staff will administer the ESAS[17] and the Montreal Cognitive Assessment (MoCA)[23] at baseline. An observational logbook or diary will be kept to record information not collected in the existing data fields. Table 1 indicates data collected at each data collection time points.

## Measures

### Edmonton Symptom Assessment Scale

The ESAS is a 10-item numeric rating scale assessing self-reported pain, fatigue, nausea, depression, anxiety, drowsiness, shortness of breath, appetite, well-being and sleep.[17] Participants or their carer will rate the severity of each symptom on a scale of 0–10 by circling a number on each scale which best describes their symptoms over the past 24 hours.[17] Higher scores indicate higher symptom burden.

### Montreal Cognitive Assessment

The MoCA is a 1-page, 30-point screening instrument that assesses cognitive domains of memory, language, executive functions, visuospatial skills, calculation, abstraction, attention, concentration and orientation.[23] A cut-off score of 26 has been shown to differentiate mild cognitive impairment or dementia from normal cognition.[23] When scoring, a 1-point adjustment is given to those with 12 or less years of formal education.[23]

### Fibion SENS

The activity tracker is the Fibion SENS device (SENS Motion, Copenhagen, Denmark),[18] which is a waterproof three-dimensional triaxial accelerometer (4.7 cm×2.2 cm×0.45 cm) previously validated for use in older adult samples.[24 25] The SENS device samples data at a rate of 12.5 Hz and an algorithm categorises data into predefined activities: lying/sitting rest; lying/sitting movement; standing; walking; sporadic walking; cycling and similar of light, moderate and vigorous intensity. Residents will be asked to continually wear the tracker if possible, or for a minimum of 16 hours per day to ensure sufficient data collection. The manufacturer recommends attaching the device to the lateral distal thigh with a medical adhesive patch approximately 10 cm above the lateral epicondyle.

### Fibion Emfit

The sleep sensor is the Fibion Emfit sensor (Emfit, Kuopio, Finland).[19] The device is a thin, flexible under-mattress device (32 cm×62 cm×0.4 cm) comprising charged polymer layers and air voids, the position of which changes with mechanical pressure to measure body movement.[26 27] The ballistic forces generated collect activity and physiological data during sleep including sleep time, sleep classes, heart rate, breathing rate, movement and bed occupancy and exits.[27]

## Phase I

### Dashboard integration

A protocol will be developed for extracting data from digital technologies (wearable activity trackers and sleep tracking sensors) and resident clinical information and medication records into the TCC-ADEPT platform. The TCC-ADEPT platform will include a dashboard with longitudinal visualisations of changes in residents' health and well-being over time (ie, physical activity, sleep quality, blood pressure, etc) using data loaded from the digital technologies and clinical data (figure 1). Medicine use (including sedative load and cumulative medicines risk calculator) and pathology records (eg, renal function) will also be uploaded to the dashboard. The platform infrastructure will be hosted on the Microsoft Azure cloud service with built-in regulatory compliance for the Health Insurance Portability and Accountability Act, and the Australian Government Information Security Manual. The resulting dashboard will be operated by the on-site pharmacist in phase II (feasibility study).

**Table 1** Data collection schedule

| Variable | Description | Data source | Data collection time points | | | |
|---|---|---|---|---|---|---|
| | | | **Baseline** | 4 weeks | 8 weeks | 12 weeks |
| Demographic | Age, gender | Leecare | ✓ | | | |
| Medical conditions | Diagnosis of medical condition | Leecare | ✓ | ✓ | ✓ | ✓ |
| Medications* | Name, strength, dose, frequency of medicine administration | Medi-Map | ✓ | ✓ | ✓ | ✓ |
| Nutritional status | Swallowing status | Leecare | ✓ | | | ✓ |
| Functional ability | Mobilisation status | Leecare | ✓ | | | ✓ |
| Cognitive function | Montreal Cognitive Assessment | Leecare | ✓ | | | ✓ |
| Falls history | Number of falls and related health outcomes | Leecare | ✓ | | | ✓ |
| Vital signs | Height, weight, blood pressure (systolic and diastolic), respirations, pulse, BGL, $SpO_2$ | Leecare | ✓ | ✓ | ✓ | ✓ |
| Pain | PainChek score severity across six domains | Leecare | ✓ | ✓ | ✓ | ✓ |
| Residential Medication Management Review (RMMR) | Dates of RMMR provision will be recorded | On-site aged care pharmacist | ✓ | ✓ | ✓ | ✓ |
| Physical activity | Time spent in light, moderate, vigorous-intensity activity | Fibion SENS device | ✓ | ✓ | ✓ | ✓ |
| Sleep | Movements during sleep, durations of light, deep and REM sleep | Fibion Emfit device | ✓ | ✓ | ✓ | ✓ |
| Edmonton Symptom Assessment Scale | Severity across 10 symptoms | Participant or third-party informant review | ✓ | ✓ | ✓ | ✓ |
| Adverse events | For example, falls, fracture, delirium, faecal impaction, incontinence | Leecare, pharmacists, aged care staff, residents | ✓ | ✓ | ✓ | ✓ |
| Incidents | Medication incidents will be recorded including date/time and severity | Leecare, pharmacists, aged care staff, residents | ✓ | ✓ | ✓ | ✓ |
| Ambulance call-outs | Date and reasons will be recorded | Leecare, pharmacists, aged care staff, residents | ✓ | ✓ | ✓ | ✓ |
| Hospital visits | Date and reason of emergency department visits or hospital admission will be recorded | Leecare, pharmacists, aged care staff, residents | ✓ | ✓ | ✓ | ✓ |
| All-cause mortality | Any deaths in the 12-week study period will be recorded | Aged care staff | | ✓ | ✓ | ✓ |

*Data will be collected and exported daily into the TCC-ADEPT platform over the 12-week study period.
BGL, blood glucose level; REM, rapid eye movement; $SpO_2$, oxygen saturation; TCC, TeleClinical Care.

## Identification of triggers for review

To determine whether a resident has experienced medicine-induced harm, the on-site pharmacists will review data from the digital technologies, residents' clinical and medicine data presented in the TCC-ADEPT dashboard. Baseline data collected within the first 2 weeks

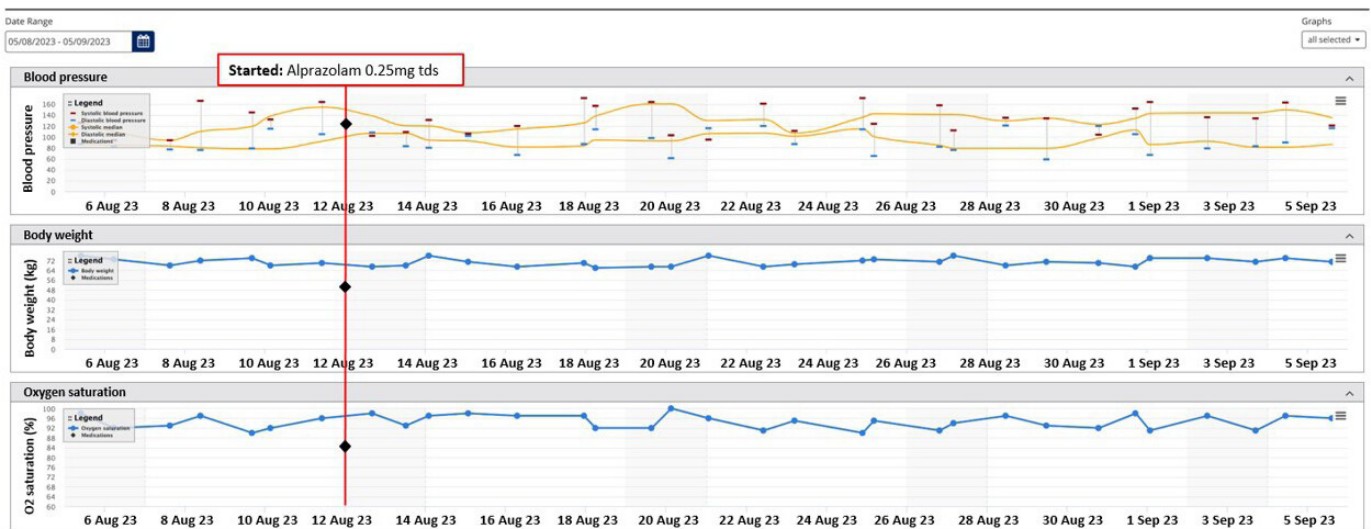

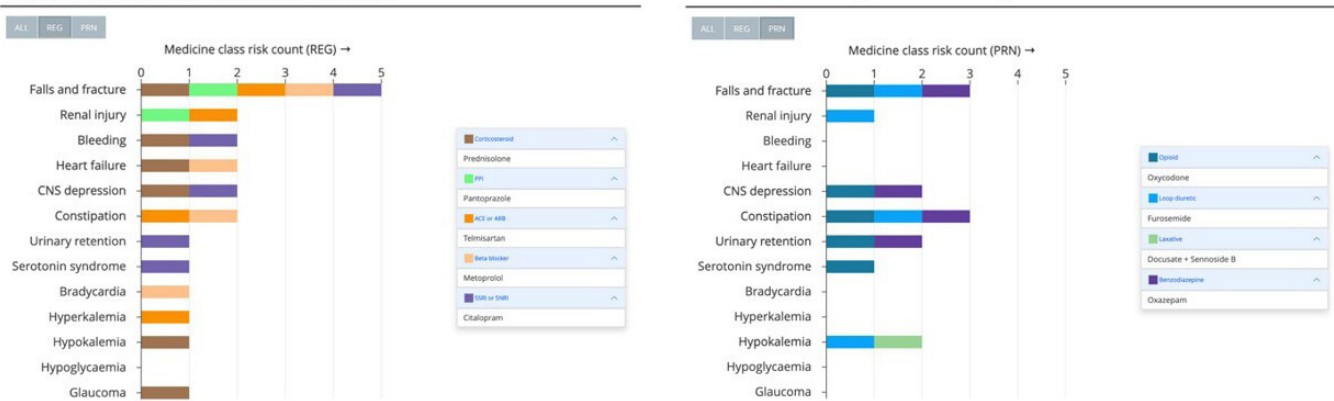

**Figure 1** (A) Example of vital signs (blood pressure, body weight and oxygen saturation) with onset of a new medication (alprazolam 0.25 mg three times a day) overlaid in the TCC-ADEPT platform. (B) Example of cumulative risks and type of adverse drug reactions displayed in the TCC-ADEPT platform. CNS, central nervous system; TCC, TeleClinical Care.

of the study will be averaged and serve as the reference point, and the dashboard will present longitudinal data and trends for each resident. To determine whether changes in health and function are medicine-induced, visualisations will be presented relative to each time point where medicines are added, ceased or modified.

In-built data analytics running an automated rules-based algorithm will be used for detection of changes in physiological or other health parameters. If and when residents' health declines beyond a certain threshold compared with baseline levels, the system will automatically flag residents that will alert the pharmacist (figure 2).

The trigger threshold will be determined based on minimum clinically significant difference where known in the literature (table 2). Where minimum clinically

significant differences are not known, a change of 10% will be considered a trigger, or in consultation with our clinicians.

## Phase II

The pre-post feasibility study will be conducted over 12 weeks. The study will be reported according to the Consolidated Standards of Reporting Trials check-list for pilot and feasibility trials.[28] The sleep sensor (Fibion Emfit)[19] will be installed in the rooms of all enrolled and consenting residents, and study staff will provide residents with an activity tracker (Fibion SENS).[18] Two weeks of baseline data will be collected from these technologies prior to the start of the pharmacist service.

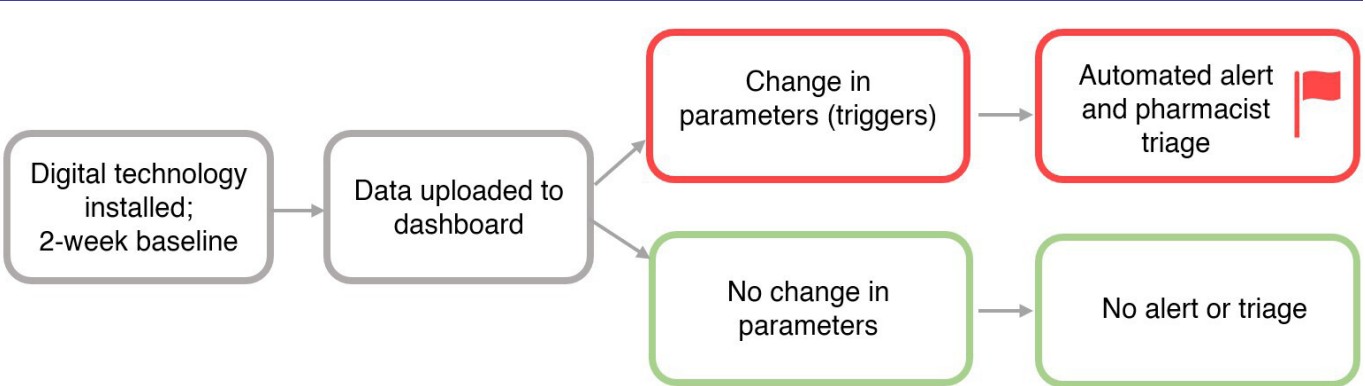

**Figure 2** Proposed service flow.

## Pharmacist service

The on-site pharmacist will monitor the participants using the TCC-ADEPT dashboard which includes data from the digital technologies as well as daily uploaded clinical and medicine data. Where clinically significant changes are detected, the resident will be automatically flagged for the pharmacist to take appropriate action (eg, continue monitoring, perform a review or initiate multidisciplinary review).

## Sample size

A sample size of 30–50 participants is recommended for pilot and feasibility studies.[29] The current study aims to recruit 30 participants from a 120-bed residential aged care facility.

## Outcomes and analysis

Study outcomes will be measured at 4-week, 8-week and 12-week postbaseline period. Health outcomes include number of medication adverse events and change in resident symptom score (ESAS); surrogate measure outcomes include change in physical activity and sleep; process measure outcomes include number of pharmacist recommendations provided, record of staff time spent during study implementation period and proportion of all pharmacists' recommendations to GPs and aged care staff that are implemented.

## Process evaluation to assess the feasibility of the pharmacist service

A mixed methods evaluation, applying the RE-AIM evaluation framework[30] will be performed to assess the feasibility of the pharmacist service. Descriptive statistics will be employed to report demographic characteristics and health information obtained from the Resident Care Assessment Record.

To determine *reach*, the number and proportion of eligible residents who were willing to participate in the study will be calculated. Information that will be collected include number of residents approached by the study staff or pharmacist and provided with the study information, number of residents who meet the inclusion criteria, number of residents who provided consent and number of participants who completed the 12-week study.

To determine *effectiveness*, changes in resident symptom score predigital and postdigital pharmacist service, as measured using the ESAS, will be assessed. Changes in medication use will be assessed, including changes to sedative and anticholinergic load predigital and postdigital pharmacist service. Changes in 24-hour movement behaviour, as measured using the activity tracker, will be assessed predigital and postdigital pharmacist service. The number of adverse events including falls (both injurious and non-injurious), fractures, delirium, faecal impaction, hospitalisation or emergency department presentation due to adverse medication events and medication incidents will be assessed. The number of adverse events experienced by residents within the 12-week predigital pharmacist service versus number of adverse events experienced during the 12-week digital pharmacist service period will be compared. Pre-post paired t-test or Wilcoxon

| Table 2 | List of triggers |
|---|---|
| | **Trigger (change from participants baseline value)** |
| Medicine data | Sedative load increase by ≥2.[13] Anticholinergic load increases ≥2.[13] Number of medicines contributing to each adverse drug reaction listed in the Veterans' MATES adverse event cumulative risk calculator ≥2.[37] |
| Resident care record | Record of common medicine-induced symptoms, for example, constipation, dry mouth, delirium. Hospital discharge or emergency department admission in the past week. |
| Activity tracker | ≥30 min in daily sedentary time.[14 38] |
| Sleep sensor | ≥10% reduction in sleep quality; ≥1 hour change in sleep time.[39 40] |
| Blood pressure | ≥10 mm Hg change in systolic blood pressure; ≥5 mm Hg in diastolic blood pressure.[41] |
| Weight | ≥5% change in weight.[42] |
| Blood glucose level | ≥10% change. |

test will be used for continuous data and McNemar's tests for categorical variables.

To assess service *adoption*, the number and proportion of organisation staff invited, that agreed to participate, will be calculated.

To assess *implementation*, the number and types of pharmacist recommendations will be recorded and evaluated. In addition, the proportion of recommendations made by the pharmacist will be compared with the number of recommendations accepted, to determine the acceptability of the digital pharmacist service by the GPs and aged care staff. Semi-structured interviews or focus groups will be conducted with the project pharmacist, GPs, aged care staff, residents and family members to understand their views of the digital pharmacist service, including facilitators and barriers to its delivery. An analysis will be undertaken of the costs associated with implementing the ADEPT-assisted pharmacist service. The resource use associated with the implementation of the ADEPT pharmacist service, including staff (aged care and health professionals) time, equipment and resources will be recorded by trial staff prospectively. Changes in pharmacist time spent reviewing residents will be measured using pharmacist's record of amount of time spent reviewing medications for residents, generating recommendations and communicating with staff, GPs and residents. Resources will be valued using appropriate unit costs drawn from organisational finance data, relevant published employment awards,[31] the Medicare Benefits Schedule (list of medical services that the Australian Government provides a rebate to assist with costs) and the Pharmaceutical Benefits Scheme (Australian Government programme that subsidises costs of prescription medications).[32 33] Where valuations are unable to be obtained from published data, we will value using market rates, using the average cost from three suppliers. Prices will be updated to the current year using the Consumer Price Index.[34] A budget impact analysis will be conducted, extrapolating the within-study assessment of costs to a wider population level.

To assess *maintenance*, interviews with senior management at the aged care facility will be conducted to understand intention to continue to implement the service poststudy funding, and whether the service aligns to the facility's mission or sustainability of business model. The capacity of the aged care organisation to implement the service (beyond the feasibility study) to identify and negate potential medicine-induced harms will be explored.

## Informed consent

Prior to the conduct of any study procedures, study staff will provide residents or an authorised representative with the participant information sheet and consent form. The purpose of the study and all study procedures will be discussed with the resident or an authorised representative. Informed written consent will be obtained by the resident or authorised representative. Participants will be assured that participation is completely voluntary and that they can withdraw from participation at any time.

## Data management

All identifying information will be kept in secure settings in the residential aged care facility, University of South Australia (QUMPRC) or University of New South Wales, accessible only by central study staff. Where necessary, electronic tablets used for data collection (under password protection), and hard copies of questionnaires, will be stored temporarily within the residential aged care facility. The electronic tablets and forms will be stored in locked cabinets within the facility (accessible only by study staff). Information containing identifiers will be removed at the end of data collection by the project lead. The QUMPRC physical and IT environment operates to in-confidence and protected levels according to Department of Defence security standards.

## Risks

The aim of the current study is to test the feasibility of using digital technologies to help pharmacists better monitor the effects of medicines that the residents are taking. There are no apparent risks to participants that would be expected as a result of the pharmacist service or the unintrusive activity and sleep monitors. It is anticipated that there may be direct benefits to the participants through a reduction in adverse events.

## Ethics and dissemination

Ethical approval has been obtained from the University of South Australia Human Research Ethics Committee (HREC205098). The study will be conducted in accordance with principles of the Declaration of Helsinki[35] and the National Statement of Ethical Conduct in Human Research.[36] During the study, any amendment to the study protocol or related documents will be submitted to the ethics committees for approval prior to implementation.

A Clinical Study Report in accordance with relevant International Council for Harmonisation of Technical Requirements for Pharmaceuticals for Human Use (ICH) guidelines will be prepared, summarising all study data and analysis. Study findings may also be published or presented at scientific meetings. All publications arising from the study will comply with recognised ethical standards concerning publications and authorships. In all instances, dissemination of the study results will include de-identified or average study data and will not identify an individual participant.

**Author affiliations**
[1]Quality Use of Medicines and Pharmacy Research Centre, University of South Australia, Adelaide, South Australia, Australia
[2]Tyree Foundation Institute of Health Engineering (IHealthE) and Graduate School of Biomedical Engineering, University of New South Wales, Sydney, New South Wales, Australia

³Tanunda Lutheran Home Inc, Tanunda, South Australia, Australia
⁴SA Pharmacy, Adelaide, South Australia, Australia
⁵UniSA Clinical and Health Sciences, University of South Australia, Adelaide, South Australia, Australia
⁶Alliance for Research in Exercise, Nutrition and Activity, University of South Australia, Adelaide, South Australia, Australia
⁷Rehabilitation, Aged and Extended Care, Flinders University, Adelaide, South Australia, Australia
⁸Quality Use of Medicines and Pharmacy Research Centre, UniSA Clinical and Health Sciences, University of South Australia, Adelaide, South Australia, Australia
⁹Pharmaceutical Society of Australia, Deakin, Australian Capital Territory, Australia
¹⁰HealthyCare Services, Welland, South Australia, Australia
¹¹School of Allied Health, The University of Western Australia, Perth, Western Australia, Australia

**Contributors** PB, TM, DD, RM, EER and RL conceived the study and obtained study funding. MB, PB, JS, DD, TM and RL assisted in the development of the study protocol. MB drafted the protocol manuscript and coordinates the day-to-day operation of the study. PB developed the TeleClinical Care dashboard for the ADEPT study based on designs and prior development by NHL and PB. All authors critically revised and approved the final manuscript.

**Funding** This project is funded by an Aged Care Research & Industry Innovation Australia (ARIIA) grant (GO000039). JS is employed by the organisation providing aged care service to the study participants. DD is supported by a Discovery Early Career Research Award (DE230101174). RL is supported by National Health and Medical Research Council Early Career Fellowship (APP1156368).

**Competing interests** None declared.

**Patient and public involvement** Patients and/or the public were involved in the design, or conduct, or reporting, or dissemination plans of this research. Refer to the 'Methods' section for further details.

**Patient consent for publication** Not applicable.

**Provenance and peer review** Not commissioned; externally peer reviewed.

**Data availability statement** No data are available. Individual participant data will not be made available due to data being collected from a small group of aged care residents located at one facility in South Australia, and the high risk of re-identification of individual participants.

**ORCID iDs**
Monique S Boord http://orcid.org/0000-0002-3033-922X
Elizabeth E Roughead http://orcid.org/0000-0002-6811-8991
Renly Lim http://orcid.org/0000-0003-4135-2523

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
