## [Reviewer comments · BMJ Open]

ARTICLE DETAILS

TITLE (PROVISIONAL)	A Digitally Enabled, Pharmacist service to detect medicine harms in residential aged care (nursing home) (ADEPT): protocol for a feasibility study.
AUTHORS	Boord, Monique; Brown, Peter; Soriano, Julian; Meola, Tahlia; Dumuid, Dorothea; Milte, Rachel; Roughead, Elizabeth; Lovell, Nigel; Stone, Helen; Whitehouse, Joseph; Janetzki, Jack; Gebreyohannes, Eyob Alemayehu; Lim, Renly

VERSION 1 – REVIEW

REVIEWER	Pace, Jessica The University of Sydney, Sydney Pharmacy School
REVIEW RETURNED	20-Nov-2023

GENERAL COMMENTS	Thank you for the opportunity to review this manuscript. This is an interesting and important study and I feel that the protocol is acceptable for publication in its current form. I look forward to reading the results.
--

REVIEWER	Bennett, Charles L University of South Carolina
REVIEW RETURNED	25-Nov-2023

GENERAL COMMENTS	nice paper.... Is it possible to add tentative study dates?
---

REVIEWER	Bergh, Sverre Innlandet Hospital Trust
REVIEW RETURNED	21-Dec-2023

GENERAL COMMENTS	Review Boord et al BMJ Open 2023 Thank you for the opportunity to review your manuscript. Using digital sensors to monitor patients' symptoms and health to predict adverse events from medication is promising and can support clinicians to take better decisions in medication prescription and doses. Patients living in long term care facilities may have cognitive decline and dementia which can reduce the ability to communicate their symptoms. Introducing digital sensors in this setting may add on the already ongoing work to reduce adverse events. The manuscript describes how digital sensors could support on-site pharmacists in their interpretation of possible adverse events. The feasibility study design should have other outcomes than larger RCT studies, where the main aim is to assess feasibility and get information to plan larger RCTs.
--

Although the manuscript generally is well written and presents information in a clear way, there are some weaknesses that I have addressed below. I have divided my comments in major and minor comments. Some of the major comments relates to the study design, and if the recruitment has started these comments may not be able to address.

Major comments:

- It is difficult to understand what the intervention is, which should be clarified for the reader (and for you?). It seems very complex, and it includes the assessment with the digital sensors, assessments with other assessment tools and clinical work up, communication to/with the general practitioner and staff at the care facility, and medication review. Please include a chapter describing exactly the intervention and what part of the intervention that is new for this project.
- You list several outcomes on page 11 (line 128-133). I would suggest choosing one primary outcome. I am also worried that some of the data that informs the pharmacists about adverse effects are also used as outcomes. Please rethink if the same data variable could be both information to start the intervention and an outcome.
- I support the use of the RE-AIM framework for the evaluation of the feasibility study. But, depending on what the intervention is the different parts of the evaluation must be revised. As an example, it seems that introducing digital sensors is the intervention, but for Reach in the RE-Aim you have chosen the proportion of eligible participants that were included (in addition to the proportion of participants that received the intervention). I can't see that your intervention is expected to increase the inclusion rate, so why is recruitment proportion part of the evaluation. Please explain?
- Further, for the Adaption part you will again calculate the proportion of eligible participants recruited. Usually, Adaption is used to describe the proportion of staff in the organization reached. Please revise.
- Maintenance of an intervention is important, and the last part of the RE-AIM. Exploring the intention to continue the intervention/service is ok, but much better would be to measure if the intervention was in use some months after the study period. Please elaborate.
- It is not clear for me if the two weeks of baseline assessment with digital sensors are before the baseline assessment with ESAS etc. Please clarify.
- It is also a concern that these two weeks should be a reference for the rest of the study. There could be incidents during these two weeks (like infections, urine retention, pain etc.) that alters the vital signs, physical activities, and sleep for the participants. How would you secure that these two weeks represent the steady state for the participants?

	 - It seems from figure 1A and B that blood pressure, body weight etc. are measured every other day, but from table 1 it seems like these vital signs are measured at baseline, 4-, 8, and 12 weeks. Please clarify. - Would you introduce a standard assessment tool for adverse effects? - Statistical plans are not described in the manuscript, and a preliminary plan for the analysis of the quantitative data must be included in the manuscript. Minor comments:  - The SPIRIT checklist does not seem to correspond with the pages in the manuscript we have received for review. - The protocol should have a date and version identifier. - The Medicare Benefits and Pharmaceutical Scheme may not be familiar for the reader and should be described. - Residential Medication Management Review should also be explained better.
--	--

VERSION 1 – AUTHOR RESPONSE

Reviewer one

1. **Thank you for the opportunity to review this manuscript. This is an interesting and important study and I feel that the protocol is acceptable for publication in its current form. I look forward to reading the results.**

We thank the Reviewer for the time taken to review our manuscript.

Reviewer two

1. **Nice paper.... Is it possible to add tentative study dates?**

Thank you for the time taken to review our manuscript. We have now included the start and end dates for the study on page 7. Lines 159-160 now read as follows:

“Recruitment and study implementation will occur from August 2023 until February 2024.”

Reviewer three

1. **Thank you for the opportunity to review your manuscript. Using digital sensors to monitor patients’ symptoms and health to predict adverse events from medication is promising and can support clinicians to take better decisions in medication prescription and doses. Patients living in long term care facilities may have cognitive decline and dementia which can reduce the ability to communicate their symptoms. Introducing digital sensors in this setting may add on the already ongoing work to reduce adverse events. The manuscript describes how digital sensors could support on-site pharmacists in their interpretation of possible adverse events. The feasibility study design should have other outcomes than larger RCT studies, where the main aim is to assess feasibility and get information to plan larger RCTs. Although the manuscript generally is well written and presents information in a clear way, there are some weaknesses that I have addressed below. I have divided my comments in major**

and minor comments. Some of the major comments relates to the study design, and if the recruitment has started these comments may not be able to address.

Thank you for the time taken to review our manuscript and provide us with feedback. We agree more conclusions should come from a feasibility study than just getting information to plan a larger RCT. When we publish the results of this feasibility study, we will describe other outcomes in detail relating to our aims. We have addressed the remaining individual concerns below.

- 2. It is difficult to understand what the intervention is, which should be clarified for the reader (and for you?). It seems very complex, and it includes the assessment with the digital sensors, assessments with other assessment tools and clinical work up, communication to/with the general practitioner and staff at the care facility, and medication review. Please include a chapter describing exactly the intervention and what part of the intervention that is new for this project.**

Thank you for the opportunity to clarify. We have revised the manuscript now to avoid the use of the term intervention (instead replaced with digital pharmacist service) as all participants recruited are receiving the digital pharmacist service. We apologise if they digital pharmacist service is not clear in the manuscript. This new service involves the pharmacist using the ADEPT-TCC platform, which is a cloud-based monitoring platform that combines clinical data from multiple sources e.g., residents aged care record (demographics, vital signs), medication data (current regular medications, PRN medications), as well as sleep data gathered from a non-invasive sleep monitor and physical activity data (collected via accelerometers) into one centralised platform for the on-site pharmacist to monitor aged care residents. The intervention is the pharmacist service, and we have included a separate section describing the service on page 13 (starting line 275).

- 3. You list several outcomes on page 11 (line 128-133). I would suggest choosing one primary outcome. I am also worried that some of the data that informs the pharmacists about adverse effects are also used as outcomes. Please rethink if the same data variable could be both information to start the intervention and an outcome.**

Thank you for your suggestions and the opportunity to clarify your concern relating to data informing the pharmacist about adverse events being used as outcomes. The data presented in the ADEPT-TCC platform monitored by the pharmacist is used to either inform Residential Medication Management Reviews (i.e. medication review by pharmacist upon referral from a residents General Practitioner) or to inform recommendations made to residents General Practitioner e.g., a change in medication dose, timing based upon the clinical data seen in TCC to avoid potential adverse events. For our pre-post outcome analysis, we extract adverse events already identified in progress notes in residents aged care record, usually written by Registered and Enrolled Nurses; this informs us how many adverse events occur during baseline, 4-, 8-, and 12-weeks and will be used for pre-post comparison. We also collect data used for our outcome analysis that is not included in the ADEPT-TCC digital pharmacist service e.g., MoCA (cognition) and ESAS (symptom burden) scores. Feasibility studies focus more on the effectiveness at the place of implementation than outcome evaluation and is conducted prior to main studies [1, 2, 3]. As this is a feasibility study, we focus on assessing feasibility of recruitment, data collection methods, potential challenges, and overall project viability, rather than establishing specific primary or secondary outcomes.

[1] National Institute for Health Research (2019). Guidance on applying for feasibility studies. <https://www.nihr.ac.uk/documents/nihr-research-for-patient-benefit-rfpb-programme-guidance-on-applying-for-feasibility-studies/20474?pr=>

[2] Oka et al (2021). Review of Feasibility Studies to Ensure Conducting the Proper Nursing Intervention Research. *International Archives of Nursing and Health Care*. <https://doi.10.23937//2469-5823/1510153>

[3] Arain et al. (2010). What is a pilot or feasibility study? A review of current practice and editorial policy. *BMC Medical Research Methodology*. <https://doi.org/10.1186/1471-2288-10-67>

- 4. I support the use of the RE-AIM framework for the evaluation of the feasibility study. But, depending on what the intervention is the different parts of the evaluation must be revised. As an example, it seems that introducing digital sensors is the intervention,**

but for Reach in the RE-Aim you have chosen the proportion of eligible participants that were included (in addition to the proportion of participants that received the intervention). I can't see that your intervention is expected to increase the inclusion rate, so why is recruitment proportion part of the evaluation. Please explain?

Apologies if this was not clear in the manuscript but the digital sensors are being employed to objectively measure physical activity and sleep that will be compared pre/post digital pharmacist service, as well as being included in our digital pharmacist service (ADEPT-TCC) for monitoring. I think our wording in this section may have introduced some confusion. In response to Reviewer 3's second comment we replaced the word 'intervention' with 'digital pharmacist service' to reflect that all participating residents received this service. We have included recruitment proportion as part of this evaluation process to align with the definition of reach listed on the RE-AIM website: "The absolute number, proportion, and representativeness of individuals who are willing to participate in a given initiative, intervention, or program, and reasons why or why not." The reach section of the manuscript now reads as follows on page 14, lines 299-302:

"Information that will be collected include number of residents approached by the study staff or pharmacist and provided with the study information, number of residents who meet the inclusion criteria, number of residents who provided consent, and number of participants who completed the 12-week study."

5. Further, for the Adaption part you will again calculate the proportion of eligible participants recruited. Usually, Adaption is used to describe the proportion of staff in the organization reached. Please revise.

Thank you for highlighting this and apologies for our misunderstanding of the adoption section of the RE-AIM framework in our current manuscript. Given this is a smaller feasibility study and is only funded for one year, only one residential aged care facility was approached that employs only one on-site pharmacist. We note that the RE-AIM website states that adoption may or may not be assessed in every project. Further, they state that in projects not directly measuring adoption, stakeholder perceptions/expectations of both setting and staff level adoption can be assessed using methods including surveys and interviews that inform planning for subsequent projects. We will be conducting interviews with staff at the residential aged care facility as well as the pharmacist to assess future adoption of this new digital pharmacist service. We have amended the section in the manuscript to reflect that adoption will assess the proportion of staff within the organisation that were invited and agreed to participate. Page 15 lines 315-316 now read as follows:

"To assess service adoption, the number and proportion of organisation staff invited, that agreed to participate, will be calculated."

6. Maintenance of an intervention is important, and the last part of the RE-AIM. Exploring the intention to continue the intervention/service is ok, but much better would be to measure if the intervention was in use some months after the study period. Please elaborate.

This is a very interesting avenue and ideally, we would have liked to measure whether the intervention is used long term. Unfortunately, this falls outside the scope of the current study due to the limited funding that we have received. We will include this as a limitation when our study results are published.

7. It is not clear for me if the two weeks of baseline assessment with digital sensors are before the baseline assessment with ESAS etc. Please clarify.

Thank you for the opportunity to clarify. At participants initial baseline assessment, study staff administered the ESAS and MoCA. The digital sensors (sleep and activity) then collected two weeks of data and other data points indicated in table 1 were retrospectively extracted from residents aged care record over that time.

8. It is also a concern that these two weeks should be a reference for the rest of the study. There could be incidents during these two weeks (like infections, urine retention, pain etc.) that alters the vital signs, physical activities, and sleep for the

participants. How would you secure that these two weeks represent the steady state for the participants?

This is a great point, and we understand the concern. Given the health status and varying comorbidities of aged care residents, we cannot guarantee a steady state for participants. To mitigate this, we averaged data collected over this two-week period to form the reference. We have made this clearer on page 12, line 246 which now reads:

“Baseline data collected within the first two weeks of the study will be averaged and serve as the reference point, and the dashboard will present longitudinal data and trends for each resident.”

9. It seems from figure 1A and B that blood pressure, body weight etc. are measured every other day, but from table 1 it seems like these vital signs are measured at baseline, 4-, 8, and 12 weeks. Please clarify.

Thank you for the opportunity to clarify this. Table 1 displays the outcomes that we are collecting at baseline, 4-, 8-, and 12- weeks for our outcome analysis (assessing pre/post changes). Figures 1A and B are displaying the interface of our digital pharmacist service (ADEPT-TCC) that the on-site pharmacist uses to monitor residents' health and the information presented here is continuous.

10. Would you introduce a standard assessment tool for adverse effects?

For the current study we are not using a standardised assessment tool for measuring adverse effects. We are using progress notes written by the aged care staff to identify any adverse events (or signs/symptoms indicative of an adverse event) that occur over the duration of the 12-week study. In future we would consider using a standard assessment for estimating the probability of adverse events. Adverse events collected for the current study are based on the Reducing Medicine-induced Deterioration and Adverse Reactions (ReMInDAR) trial [4] e.g., falls (injurious and non-injurious, fractures, delirium, faecal impaction, hospitalisation.

[4] Roughead et al. (2022). Effect of an ongoing pharmacist service to reduce medicine-induced deterioration and adverse reactions in aged-care facilities (nursing homes): A multicentre, randomised controlled trial (the ReMInDAR trial). *Age and Ageing*. <https://doi.org/10.1093/ageing/afac092>

11. Statistical plans are not described in the manuscript, and a preliminary plan for the analysis of the quantitative data must be included in the manuscript.

Our preliminary analysis plan is located on page 18 under the section “Outcomes and analysis”. We recognise the statistical plan for pre-post comparisons was not clear and have added the following to the effectiveness section of the RE-AIM framework on page 15 line 313-314:

“Pre-post paired t-test or Wilcoxon tests will be utilised for continuous data and McNemar’s tests for categorical variables.”

12. The SPIRIT checklist does not seem to correspond with the pages in the manuscript we have received for review.

Apologies that the pages listed in the checklist did not align with those in the manuscript received. I believed what has happened is that when ScholarOne Manuscripts combines our submitted documents into a proof in, a blank page is added between the first page of the review document and when the manuscript starts, hence the displaced page numbers.

13. The protocol should have a date and version identifier.

All ADEPT protocols used in executing the study and data collection have a date and version identifier. We did not however include a date and version identifier in this manuscript for publication. If the editor requires this, we would be happy to provide one.

14. The Medicare Benefits and Pharmaceutical Scheme may not be familiar for the reader and should be described.

Thank you for your suggestion. We have revised the manuscript to explain the Medicare Benefits Schedule and the Pharmaceutical Benefits Scheme. Page 15 lines 332-334 now reads as follows:

“...the Medicare Benefits Schedule (list of medical services that the Australian Government provides a rebate to assist with costs) and the Pharmaceutical Benefits Scheme (Australian Government program that subsidises costs of prescription medications).”

15. Residential Medication Management Review should also be explained better.

Thank you for your suggestion. We have revised the manuscript to explain Residential medication Management Review. Page 4 lines 87-88 now reads as follows:

“Existing services are limited to government funded residential medication management review (comprehensive medication reviews performed by accredited pharmacists in residential aged care) and quality use of medicine services.”

Reviewer_response_v323012023

VERSION 2 – REVIEW

REVIEWER	Pace, Jessica The University of Sydney, Sydney Pharmacy School
REVIEW RETURNED	29-Jan-2024

GENERAL COMMENTS	Thank you for your efforts revising this manuscript. All reviewer comments have been addressed and I now deem it suitable for publication.
--

REVIEWER	Bergh, Sverre Innlandet Hospital Trust
REVIEW RETURNED	25-Jan-2024

GENERAL COMMENTS	Dear authors. Thank you for addressing my comments and concerns in the first review process. My understanding is that the manuscript has improved and better describes the ongoing datacollection. I wish you good luck with the study and I am looking forward to learn from your findings.
--